# Construct Validity of a New Health Assessment Questionnaire for the National Screening Program of Older Adults in Japan: The SONIC Study

**DOI:** 10.3390/ijerph191610330

**Published:** 2022-08-19

**Authors:** Tatsuro Ishizaki, Yukie Masui, Takeshi Nakagawa, Yuko Yoshida, Yoshiko L. Ishioka, Noriko Hori, Hiroki Inagaki, Kae Ito, Madoka Ogawa, Mai Kabayama, Kei Kamide, Kazunori Ikebe, Yasumichi Arai, Yasuyuki Gondo

**Affiliations:** 1Tokyo Metropolitan Institute of Gerontology, Tokyo 173-0015, Japan; 2National Center for Geriatrics and Gerontology, Obu 474-8511, Japan; 3Jindal School of Liberal Arts and Humanities, O.P. Jindal Global University, Sonipat 131001, Haryana, India; 4Clinical Thanatology and Geriatric Behavioral Science, Graduate School of Human Sciences, Osaka University, Osaka 565-0871, Japan; 5Division of Health Sciences, Graduate School of Medicine, Osaka University, Osaka 565-0871, Japan; 6Department of Prosthodontics, Gerodontology and Oral Rehabilitation, Graduate School of Dentistry, Osaka University, Osaka 565-0871, Japan; 7Center for Supercentenarian Medical Research, Keio University School of Medicine, Tokyo 160-8582, Japan

**Keywords:** construct validity, questionnaire, national screening program, older adults, Japan

## Abstract

The Japanese government has implemented a new screening program to promote measures to avoid worsening lifestyle-related diseases and frailty among the older population. In this effort, the government formulated a new health assessment questionnaire for the screening program of old-old adults aged ≥75 years. The questionnaire comprises 15 items, of which 12 address frailty, two address general health status, and one addresses smoking habits. This study examined the construct validity of this questionnaire, using the explanatory factor analysis (EFA) and confirmatory factor analysis (CFA). The data used in this study were drawn from a mail-in survey conducted in 2020 as part of the Septuagenarians, Octogenarians, Nonagenarians Investigation with Centenarians study. A total of 1576 respondents (range, 78–99 years of age) were included in the study. Although the EFA did not show an interpretable factor structure of the questionnaire with 15 items, the CFA using only 12 frailty-related items showed the goodness of fit for a higher-order factor “frailty”, and the five frailty-related sub-factors model was acceptable. These results suggest that the total score of the 12 frailty-related items in the questionnaire can be used as an indicator of the degree of “frailty”.

## 1. Introduction

In April 2008, the Japanese government implemented a nationwide annual screening program to identify the individuals who were at risk of metabolic syndrome and to provide health guidance to help reduce such a risk for all residents aged 40–74 years [1]. The annual screening program was extended to those aged 75 years and above (old-old adults). As most old-old adults have been undergoing drug therapy for their chronic conditions, such as hypertension, dyslipidemia, coronary heart disease, and osteoarthritis, and have undergone screening at the time [2,3,4], we believe that the annual screening program was of poor value for the old-old adults who were already receiving treatment for such diseases. In April 2016, the Japanese government amended the health policy for old-old adults to implement a new screening program that promotes measures to avoid the worsening of lifestyle-related diseases and frailty in each municipality [5]. Frailty is a state in which the standby capacity of the body decreases because of aging; older adults with frailty are vulnerable to various stresses, and their degree of independence in functioning is reduced [6]. As advanced frailty interferes with independence in performing the activities of daily living and increases the risk of disability, it is necessary for older adults to assess their health conditions comprehensively, to avoid frailty and worsening chronic diseases.

In April 2020, the national government formulated a new health assessment questionnaire for the annual screening program of old-old adults and asked each municipality to use it (Table 1) [7].

This questionnaire comprises 15 items, including two items on general health status, such as self-rated health and life satisfaction, 12 items on multidimensional aspects of frailty, such as physical function, nutritional status, oral function, cognitive function, and social participation, and one item on smoking habits. Because this questionnaire is aimed at providing a comprehensive assessment of the health status of older adults using each of the 15 items individually, the expert committee that developed this questionnaire did not initially envision that the items would be scored to identify the older adults with health risks [8]. However, some of the public health nurses in charge of community health expressed demands to assess the priority of the need for health guidance among the community-dwelling old-old adults by calculating the total score on this questionnaire.

A recent study examined the construct validity of the questionnaire using all 15 items [9]. This study had a few limitations. First, the authors assumed six factors were extracted from the explanatory factor analysis (EFA), including nine with factor loadings <0.4, which were rather small to the point that they should have been excluded [10]. Second, because the authors did not show the results of the tests for the statistical significance of the factor loadings in the confirmatory factor analysis (CFA), we were unable to statistically and logically interpret as independent the meanings of the sub-factors shown in the study. As the health assessment questionnaire for old-old adults was not developed with the intention of using all of the 15 items as a rating scale in the first place, we believe that the construct validity of the questionnaire does not need to be examined using all of the 15 items and should have been evaluated in a statistically appropriate manner. The current study aimed to re-examine the construct validity of this questionnaire and discuss its applicability as a comprehensive health assessment scale for old-old adults.

## 2. Methods

### 2.1. Study Design and Participants

The data used in this study were drawn from a mail-in survey conducted in August and September 2020 in the Septuagenarians, Octogenarians, Nonagenarians Investigation with Centenarians (SONIC) study [11]. The SONIC study used a narrow age-range cohort design: all of the residents in four areas (urban and nonurban) in the Tokyo Metropolitan and Hyogo Prefectures; aged 69–71 years (70s group); 79–81 years (80s group); and 89–91 years (90s group) in the initial survey, were recruited for participation in an in-venue survey. The follow-up with the participants of the in-venue survey was conducted once every 3 or 4 years. The 70s group was recruited for the initial survey in 2010. The additional participants were recruited during the first follow-up survey conducted in 2013. The 80s group was recruited for the initial survey in 2011, and additional participants were recruited during the first follow-up survey in 2014. The 90s group was recruited for the initial survey in 2012, and additional participants were recruited during the first and second follow-up surveys, conducted in 2015 and 2018, respectively. The total number of participants at the time of the baseline survey was 3346 (1592 men and 1754 women; 70s = 1129; 80s = 1234; 90s = 883). For the present mail-in survey, a letter of request and questionnaire were sent via postal mail to 2649 people (1230 men and 1419 women; 70s = 1053; 80s = 961; 90s = 635), excluding 693 dropouts from the original list of 3346 participants, due to death, institutionalization, falling in a state requiring nursing care, unknown new address, refusal to participate further, and other reasons.

### 2.2. The Health Assessment Questionnaire for Old-Old Adults

A health assessment questionnaire for the old-old adults (Table 1) [7] was used in this study. Although this was self-administered, when the respondents could not respond by themselves or answer the questions, a family member or a third person who knew them was allowed to fill it on their behalf. The responses to the individual items were converted into scores in the following manner. For Question 1, a score scale of 0 “*excellent*” to 4 points “*very poor*” in 1-point increments was used for each option. For Question 2, a score scale of 0 “*satisfied*” to 3 points “*dissatisfied*” in 1-point increments was used for each option. For Question 12, “*Yes*” was scored 2, “*No*” was scored 0, and “*I quit*” was scored 1. For questions 3, 9, 13, 14, and 15, “*Yes*” was scored 0, and “*No*” was scored 1. For questions 4–8, 10, and 11, “*Yes*” was scored 1, and “*No*” was 0.

### 2.3. Basic Characteristics of the Participants

To identify the basic characteristics of the participants, we collected data on sex, age, educational attainment, living arrangements, instrumental activities of daily living (IADL), and eligibility for public long-term care (LTC) insurance benefits. The items of the IADL included being able to use public transportation by oneself, shop for daily necessities, prepare meals, pay bills, and manage one’s own bank account [12]. For each item, a score of 1 was assigned for “yes” and 0 for “no”, and the total score for the five items was calculated. The higher the score, the more independent the participant was in the IADL. Regarding eligibility for LTC insurance benefits, all of the residents aged ≥40 years in Japan are mandatorily enrolled in the public LTC insurance system. The municipality assesses whether an insured person is eligible for insurance benefits based on an evaluation of the person’s current physical and cognitive status [13]. When an insured person is certified as having LTC needs by the local government, they are categorized into either two or five levels of support or LTC needs (Levels 1 to 5).

### 2.4. Statistical Analysis

The factor structure of the health assessment questionnaire items for the old-old adults was examined using EFA and CFA. The principal factor method was used for factor extraction from the 15 items of the questionnaire, with Promax rotation in the EFA. We used both the Kaiser–Guttman test and the scree test to determine the number of factors to be extracted in the EFA [14]. We decided that any item with a factor loading of ≥0.4 was interpretable for each factor in the EFA [10]. Because the health assessment questionnaire for old-old adults includes 12 items to assess the multidimensional aspects of frailty, such as physical, cognitive, and social aspects, we then used CFA to examine whether the 12 frailty-related items in this questionnaire fit the multidimensional model of frailty [15]. Recent review articles have described that an assessment tool for frailty should include physical, psychological, cognitive, and social dimensions [16,17], and the physical domain includes three factors: physical function, nutritional status [15], and oral health status [18]. We assume that, behind these five frailty-related domains, there is a concept of multidimensional frailty that affects each of the five domains. Therefore, we assumed that the model of frailty for this CFA had five sub-factors as the primary factors: physical function (Q7, Q8, Q9, and Q13); nutritional status (Q3 and Q6); oral function (Q4 and Q5); cognitive function (Q10 and Q11); and social aspects (Q14 and Q15); with the secondary factor of “a comprehensive concept of frailty” on top of these five sub-factors. In the CFA, using structural equation modeling (SEM), the parameter estimates in the model were led by the maximum likelihood method [19], using only 12 frailty-related items. The χ^2^ score, goodness of fit index (GFI), adjusted GFI (AGFI), root mean square error of approximation (RMSEA), and Akaike’s information criterion (AIC) were used to assess the model’s goodness of fit for the CFA. We also calculated the Cronbach’s alpha coefficient using these 12 items. A two-tailed *p*-value < 0.05 was considered statistically significant in the CFA. IBM SPSS Statistics ver.25 was used for the fundamental statistical analysis and EFA, and IBM SPSS Amos ver.25 was used for the CFA.

## 3. Results

### 3.1. Characteristics of the Participants

Among the 2653 people, 1783 responded to the questionnaire (response rate, 67.2%). A total of 1576 (88.4%) respondents who provided answers to all 15 items of the health assessment questionnaire were included in the analysis. The respondents’ basic characteristics are listed in Table 2.

The men accounted for 47.1% of the total study population. The mean age was 85.6 years (standard deviation 5.9), and the age range was 78–99 years. Of the total, 87.8% responded on their own, 6.5% responded through a proxy, 1.9% relied on a third person, and 3.9% had an unknown response status.

### 3.2. EFA

Table 3 shows the mean values for each item stratified by sex and age group.

First, to investigate the factor structure of the health assessment questionnaire for old-old adults, we conducted an EFA using the 15 items identified using the principal factor method. Based on the Kaiser–Guttman test, we extracted five factors with eigenvalues of one or more. Next, the eigenvalues of the identified factors were plotted on a scree plot (Figure 1). Because the shape of the scree plot showed a smooth decrease after the fifth factor, we determined that there were five factors in this EFA.

The proportion of variance accounted for by these five factors was 25.1%. The principal factor method was used to identify the factor loadings of each item by setting the number of factors to five. Promax rotation was performed to obtain the factor pattern matrix. Table 4 shows the factor loading of each item after factor rotation, and the estimated communality after factor identification. The factor loadings were interpretable with respect to the items of both Factor 1 (general health assessment: self-rated health status; life satisfaction) and Factor 4 (cognitive function: always asking the same thing; not remembering today’s date). However, the other factors were difficult to interpret because Factor 3 included various aspects, such as oral function, weight loss, and physical function, and Factor 5 included social relationships, nutritional status, and smoking habits. In addition, Factor 2 was difficult to interpret because there was only one item with a factor loading of ≥0.4. Although the estimated communality of items Q1 and Q2 were the highest, at 0.72 and 0.40, respectively, the communality of the eight items was approximately 0.2. Specifically, the estimated communality of item Q12 “smoking habits“ was the lowest (0.02). These results indicate that the questionnaire with 15 items was not explained by the five factors, and the EFA using 15 items did not extract interpretable factor structures.

### 3.3. CFA

The next step was the CFA, using only 12 frailty-related items. These items were developed as representatives of the five frailty-related domains: physical function (Q7, Q8, Q9, and Q13); nutritional status (Q3 and Q6); oral function (Q4 and Q5); cognitive function (Q10 and Q11); and social aspects (Q14 and Q15). CFA assumed that the model had a higher-order factor (frailty) above these five frailty sub-factors. Figure 2 shows that all of the estimated factor loadings on the 12 items were 0.34 or more, except for one item (Q 15), and all of them were statistically significant (*p* < 0.05).

Each item had a factor loading at almost the same level as initially assumed. The factor loadings of the five sub-factors from the higher-order factor were sufficiently high (i.e., 0.46 or more), and all of them were statistically significant (*p* < 0.05). The goodness of fit of the model was sufficiently, with χ^2^ (49) = 156.2 (*p* < 0.001), with GFI, AGFI, RMSEA, AIC being 0.983, 0.973, 0.037, and 214.2, respectively. The Cronbach’s alpha coefficient for the 12 items was 0.569.

## 4. Discussion

The responses to the new health assessment questionnaire for old-old adults were obtained from community-dwelling older adults through a mail-in survey that covered the participants in the SONIC study. The factor structure of the questionnaire was then examined. The EFA revealed that an interpretable factor structure was not obtained through the analysis of all 15 items. In contrast, the results from the CFA showed that there were five sub-factors: physical function; nutritional status; oral function; cognitive function; and social aspects, indicating the goodness of fit of the model with the higher-order factor of “frailty” on top of these five sub-factors. In addition, the goodness of fit of the model that assumes frailty as the secondary factor may indicate that the total score of the 12 frailty-related items in the health assessment questionnaire for old-old adults can be used as an indicator of the degree of “frailty”.

This study indicated that the total score of the 12 frailty-related items in the questionnaire showed the possibility of screening older adults with frailty. This questionnaire comprehensively covered the physical, cognitive, psychological, and social aspects of frailty. Although there are several validated frailty measurement tools [20], most of them have focused on the physical aspects of frailty. In addition to this questionnaire, only some comprehensively assessed the physical, psychological, and social aspects of frailty, such as the Groningen Frailty Indicator [21,22,23] from Germany, the Edmonton Frail Scale [24,25] from Canada, the Tilburg Frailty Indicator [26,27,28] from The Netherlands, and the Kihon Checklist [29,30] and Kaigo Yobo Checklist [31] from Japan. These comprehensive frailty assessment instruments were reported to examine convergent validity and/or concurrent validity and to compare their predictive validity: a comparison of predictive validity for disability between the Groningen Frailty Indicator and the Tilburg Frailty Indicator [32]; a comparison of predictive value for identifying frailty between the Groningen Frailty Indicator and the Tilburg Frailty Indicator [33]; an examination of the convergent validity between the electronic frailty index and the Edmonton Frail Scale [34]; a comparison of diagnostic test accuracy for identifying frailty among the Groningen Frailty Indicator, the Edmonton Frail Scale, and the Kihon Checklist [35]; a comparison of predictive value for frailty between the Kihon Checklist and the Kaigo Yobo Checklist [36]; an examination of convergent validity between the Kihon Checklist (Portuguese version) and the Edmonton Frail Scale [37].

Compared with the two existing Japanese checklists for comprehensive frailty risk assessment, the health assessment questionnaire for the old-old adults has at least two unique features. First, the questionnaire was used across Japan: 91.9% (1600/1741) of the municipalities used it in 2021 [38], and 93.9% (1503/1600) of them used it in the course of the annual screening program for old-old adults. The national government expects all of the municipalities to use the questionnaire within a few years. Second, the data from the questionnaire are routinely registered in the national health insurance database system, known as “the KDB system” [39], which links the data from the national screening program and the claims data from health and LTC insurance on an individual basis. Using the data registered in the KDB system, such as blood test data at the time of the screening program, and treatment and prescription status, each municipality can provide health guidance to those at risk of frailty to prevent them from becoming frail.

We limited CFA to 12 frailty-related items out of the 15 items in the questionnaire and excluded the other 3 (subjective health, life satisfaction, and smoking habits). However, these 3 items, and the other 12, are important for the sake of the comprehensive health assessment of old-old adults living in the community. As is true for all of the items in the questionnaire for old-old adults, it is not appropriate in health guidance for them to focus only on the items with unfavorable responses and to focus too heavily on guidance for the shift to favorable health behaviors. What is important is for the health guidance instructor to acknowledge that there are favorable health behaviors that are currently being implemented and support their continuation [5]. For the items that received unfavorable responses, it is important to consider the background and causes for such unfavorable conditions, while taking into account the subject’s injuries and psychosocial conditions, and to provide health guidance that addresses the issues while taking care not to cause physical and/or cognitive impairment.

This study has at least two limitations. First, the alpha coefficient of the total score of the 12 items in the health assessment questionnaire was relatively low at 0.57, indicating a low internal consistency. We consider the reason for the low internal consistency to be because the five domains measure different aspects of frailty, which are likely to be diverse and have high variance, namely physical, oral, nutritional, cognitive, and social aspects. Therefore, it is within our expectation that the within-subject consistency among the 12 items was limited to some extent. Second, because the analyzable subjects in this study were survivors of the SONIC study, they may be healthier and have better health behaviors than the general population who participate in the annual health screening program in Japan. We believe that the construct validity of this questionnaire needs to be re-examined using the questionnaire data registered in the KDB system, which are from old-old adults who underwent health screening programs in their communities.

## 5. Conclusions

We propose that the total score of the 12 frailty-related items in the health assessment questionnaire for old-old adults can be used as an indicator of the degree of “frailty”. We expect that health professionals working in municipalities will be able to use the total score of the 12 frailty-related items out of the 15 items of the questionnaire to identify those who are eligible for health guidance aimed at preventing frailty among community-dwelling older adults. Future research should examine the criterion validity of the health assessment questionnaire for old-old adults, and the extent to which this questionnaire correlates with actual frailty status among community-dwelling old-old adults.

## Figures and Tables

**Figure 1 ijerph-19-10330-f001:**
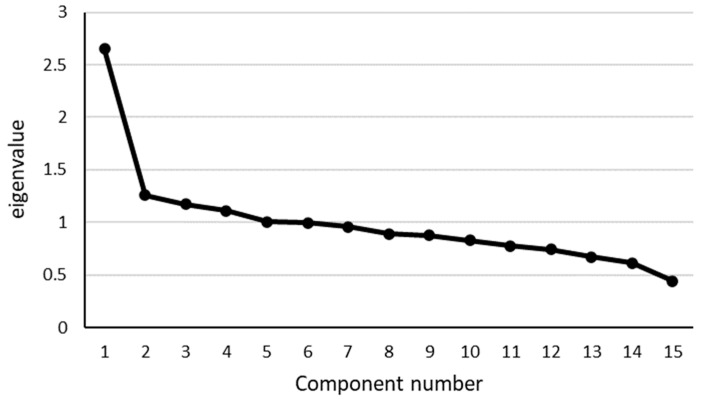
A scree plot of the health assessment questionnaire for old-old adults (15 items) in the EFA.

**Figure 2 ijerph-19-10330-f002:**
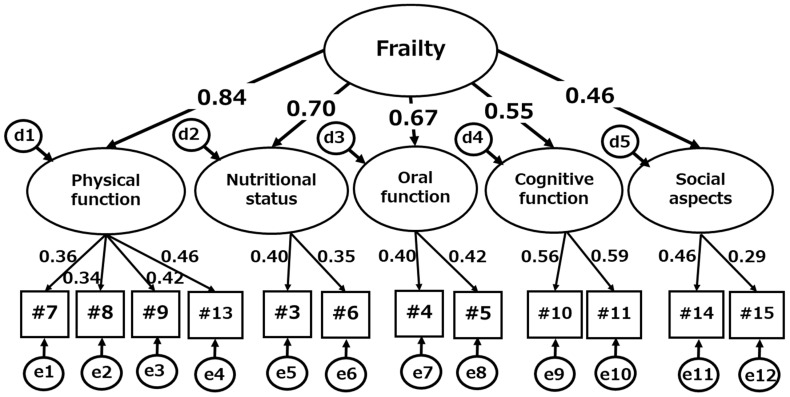
Factor structure and loadings on the 12 frailty-related items of the health assessment questionnaire for old-old adults in the CFA using SEM. χ^2^ (49) = 156.2 (*p* < 0.001), GFI = 0.983, AGFI = 0.973, RMSEA = 0.037, AIC = 214.2. All factor loadings were statistically significant (*p* < 0.05). #3 eating habits; #4 masticatory function; #5 swallowing function; #6 weight loss; #7 walking speed; #8 experiences of falls; #9 exercise habits; #10 memory loss; #11 date disorientation; #13 going out; #14 interaction with others; and #15 health-related consultation.

**Table 1 ijerph-19-10330-t001:** The health assessment questionnaire for old-old adults.

Domain	Item	Score
Health condition	1. How is your health condition?	Excellent = 0; good = 1; fair = 2; poor = 3; very poor = 4
Mental health	2. Are you satisfied with your daily life?	Satisfied = 0; moderately satisfied = 1; moderately dissatisfied = 2; dissatisfied = 3
Eating behavior	3. Do you eat three meals a day?	Yes = 0; no = 1
Oral function	4. Do you have any difficulties eating tough foods when compared to 6 months ago?	No = 0; yes = 1
	5. Have you choked on your tea or soup recently?	No = 0; yes = 1
Bodyweight loss	6. Have you lost 2 kg or more in the past 6 months?	No = 0; yes = 1
Physical function and falls	7. Do you think you walk slower than before?	No = 0; yes = 1
	8. Have you experienced a fall in the past year?	No = 0; yes = 1
	9. Do you go for a walk for your health at least once a week?	Yes = 0; no = 1
Cognitive function	10. Do your family or friends point out your memory loss? (e.g., “You ask the same questions over and over again.”)	No = 0; yes = 1
	11. Do you find yourself not knowing today’s date?	No = 0; yes = 1
Smoking	12. Do you smoke?	Yes = 2; no = 0; I quit = 1
Social participation and support	13. Do you go out at least once a week?	Yes = 0; no = 1
	14. Do you maintain regular communications with your family and friends?	Yes = 0; no = 1
	15. When you feel ill, do you have anyone to reach out/talk to?	Yes = 0; no = 1

Note: The scores shown in Satake and Arai [7] were modified by the authors.

**Table 2 ijerph-19-10330-t002:** Basic characteristics of the analyzable respondents.

	Total	Men	Women
	*n =* 1576	*n =* 742	*n =* 834
Mean age (standard deviation) (years)	85.6 (5.9)	85.7 (5.9)	85.4 (5.9)
Educational attainment (years) (mean (SD))	11.6 (2.8)	12.1 (3.1)	11.2 (2.3)
Place of living (% home)	95.1%	95.8%	94.4%
Living arrangement (% living alone)	26.2%	15.1%	36.1%
LTC need certification (% LTC need Level 1 or more)	10.6%	10.0%	11.0%
Instrumental activities of daily living (points) (mean (SD))	4.6 (1.3)	4.3 (1.3)	4.4 (1.3)

Note: SD denotes standard deviation; LTC denotes long-term care.

**Table 3 ijerph-19-10330-t003:** Mean and standard deviation of each item in the health assessment questionnaire for old-old adults with 15 items by sex and age group.

	Total (*n =* 1576)	Men (*n =* 742)	Women (*n =* 834)	70s Group (*n =* 733)	80s Group (*n =* 509)	90s Group (*n =* 334)
1. How is your health condition?	1.61 (0.98)	1.56 (1.00)	1.65 (0.96)	1.50 (0.98)	1.70 (0.95)	1.72 (1.00)
2. Are you satisfied with your daily life?	0.82 (0.70)	0.80 (0.69)	0.84 (0.72)	0.85 (0.68)	0.79 (0.72)	0.78 (0.74)
3. Do you eat three meals a day?	0.06 (0.24)	0.05 (0.22)	0.07 (0.25)	0.04 (0.19)	0.07 (0.26)	0.10 (0.30)
4. Do you have any difficulties eating tough foods when compared to 6 months ago?	0.32 (0.47)	0.28 (0.45)	0.35 (0.48)	0.24 (0.43)	0.36 (0.48)	0.42 (0.49)
5. Have you choked on your tea or soup recently?	0.22 (0.42)	0.20 (0.40)	0.25 (0.43)	0.21 (0.41)	0.22 (0.41)	0.27 (0.44)
6. Have you lost 2 kg or more in the past 6 months?	0.17 (0.37)	0.17 (0.38)	0.16 (0.37)	0.13 (0.34)	0.19 (0.39)	0.21 (0.41)
7. Do you think you walk slower than before?	0.74 (0.44)	0.73 (0.44)	0.74 (0.44)	0.64 (0.48)	0.79 (0.41)	0.86 (0.35)
8. Have you experienced a fall in the past year?	0.25 (0.44)	0.25 (0.43)	0.26 (0.44)	0.19 (0.39)	0.26 (0.44)	0.38 (0.49)
9. Do you go for a walk for your health at least once a week?	0.43 (0.50)	0.42 (0.50)	0.43 (0.50)	0.34 (0.48)	0.47 (0.50)	0.55 (0.50)
10. Do your family or friends point out your memory loss? (e.g., “You ask the same questions over and over again.”)	0.17 (0.38)	0.17 (0.38)	0.17 (0.38)	0.10 (0.30)	0.20 (0.40)	0.30 (0.46)
11. Do you find yourself not knowing today’s date?	0.31 (0.46)	0.30 (0.46)	0.33 (0.47)	0.24 (0.43)	0.33 (0.47)	0.44 (0.50)
12. Do you smoke?	0.25 (0.51)	0.46 (0.60)	0.06 (0.31)	0.30 (0.57)	0.17 (0.42)	0.25 (0.48)
13. Do you go out at least once a week?	0.17 (0.38)	0.16 (0.37)	0.18 (0.39)	0.09 (0.29)	0.20 (0.41)	0.28 (0.45)
14. Do you maintain regular communications with your family and friends?	0.06 (0.25)	0.09 (0.28)	0.04 (0.21)	0.05 (0.21)	0.07 (0.25)	0.10 (0.30)
15. When you feel ill, do you have anyone to reach out/talk to?	0.06 (0.23)	0.07 (0.25)	0.05 (0.22)	0.06 (0.25)	0.06 (0.24)	0.04 (0.20)

Note: Values are mean (standard deviation).

**Table 4 ijerph-19-10330-t004:** Factor loadings and estimated communality in the EFA using the principal factor method with Promax rotation of the health assessment questionnaire for old-old adults (15 items).

Items	F1	F2	F3	F4	F5	Estimated Communality
1. How is your health condition? (reversed)	**0.80**	0.07	0.07	−0.03	−0.03	0.72
2. Are you satisfied with your daily life? (reversed)	**0.65**	−0.03	−0.06	0.05	0.05	0.40
13. Do you go out at least once a week? (reversed)	0.00	**0.71**	−0.04	−0.03	−0.06	0.44
9. Do you go for a walk for your health at least once a week? (reversed)	0.05	0.39	0.02	0.00	0.00	0.18
4. Do you have any difficulties eating tough foods when compared to 6 months ago?	−0.04	−0.02	**0.44**	−0.03	−0.03	0.15
5. Have you choked on your tea or soup recently?	−0.04	0.03	0.36	0.03	−0.05	0.13
6. Have you lost 2 kg or more in the past six months?	0.08	−0.13	0.31	−0.03	0.13	0.13
7. Do you think you walk slower than before?	0.11	0.04	0.28	0.02	0.00	0.15
8. Have you experienced a fall in the past year?	0.01	0.09	0.21	0.08	0.02	0.11
10. Do your family or friends point out your memory loss? (e.g., “You ask the same questions over and over again.”)	−0.02	0.00	0.04	**0.63**	−0.10	0.40
11. Do you find yourself not knowing today’s date?	0.05	−0.03	−0.03	**0.54**	0.07	0.30
15. When you feel ill, do you have someone to reach out/talk to? (reversed)	0.08	−0.06	−0.11	0.00	**0.53**	0.27
14. Do you maintain regular communications with your family and friends? (reversed)	−0.10	0.19	0.05	0.04	**0.42**	0.27
3. Do you eat three meals a day? (reversed)	−0.02	0.03	0.16	−0.01	0.23	0.11
12. Do you smoke?	0.00	−0.07	0.03	−0.05	0.15	0.02

Bolded factor loading: absolute value of ≥0.4.

## Data Availability

The data supporting the findings of this study are available from the corresponding author upon reasonable request.

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
