# Peer review of "Construct Validity of a New Health Assessment Questionnaire for the National Screening Program of Older Adults in Japan: The SONIC Study"

_ijerph, 2022, doi:10.3390/ijerph191610330_

Round 1

Reviewer 1 Report

Dear authors, thank you for your work.

The current study aimed to re-examine the construct validity of the questionnaire and discuss whether the questionnaire is applicable as a comprehensive health assessment scale for old-old adults.

It is very important to have a good working tool for the evaluation of health conditions (it would be great to stress it more), but it is not easy to create it and evaluate it as well (as we can see from this paper).

In some parts the paper lacked clarity, and could be presented in a better-structured manner. It would be desirable to reformulate these paragraphs, they are not clearly explained, describe them precisely, please:(34-38 lines): The explanatory factor analysis of the 15 items revealed that the interpretable factor structure was not obtained. However, the goodness of fit of the confirmatory factor analysis for a higher-order factor model using 12 frailty-related items and 5 frailty-related subfactors, and a higher-order factor "frailty" was favorable. The results suggest that the 5 frailty-related subfactors were interpreted as statistically independent areas and the total score of the 12 frailty-related items in the health assessment questionnaire for old-old adults was statistically applicable as a score of frailty. (270-276 lines) We propose that the 12 frailty-related items in the health assessment questionnaire  for old-old adults are statistically separate descriptive assessments of frailty, as opposed  to the total score of all 15 items, and that the total score of the 12 frailty-related items in  the questionnaire is statistically applicable as the score of frailty. In future research, we  must examine the criterion validity of the health assessment questionnaire for old-old  adults and the extent to which this questionnaire correlates with actual frailty status  among community-dwelling old-old adults.

Discussion and comparison with other countries'  questionnaires about similar topics are very short (223-224 lines), please prolong it.

What are the strengths of the study?

The smaller part of the literature sources are from the last 5 years, and only 5 are from different countries than Japan, sometimes a year is not in bold, in the last source (349 line) year is missing...

Author Response

Responses to the Reviewer 1

 First of all, we appreciate your important and helpful comments on our manuscript. We have indicated in the red text where we have made revisions and underlined where we had made revisions in response to your comments/suggestions. Apart from the comments from the two reviewers, values of the estimated communality in Table 4 and the order of the subfactors in Figure 2 have been revised.

Comments to the Authors #1

In some parts the paper lacked clarity, and could be presented in a better-structured manner. It would be desirable to reformulate these paragraphs, they are not clearly explained, describe them precisely, please:

(34[33]-38 lines) The explanatory factor analysis of the 15 items revealed that the interpretable factor structure was not obtained. However, the goodness of fit of the confirmatory factor analysis for a higher-order factor model using 12 frailty-related items and 5 frailty-related subfactors, and a higher-order factor "frailty" was favorable. The results suggest that the 5 frailty-related subfactors were interpreted as statistically independent areas and the total score of the 12 frailty-related items in the health assessment questionnaire for old-old adults was statistically applicable as a score of frailty.

(270-276 lines) We propose that the 12 frailty-related items in the health assessment questionnaire for old-old adults are statistically separate descriptive assessments of frailty, as opposed to the total score of all 15 items, and that the total score of the 12 frailty-related items in the questionnaire is statistically applicable as the score of frailty. In future research, we must examine the criterion validity of the health assessment questionnaire for old-old adults and the extent to which this questionnaire correlates with actual frailty status among community-dwelling old-old adults.

Response

Thank you for your comment. As suggested, we have revised lines 34-38 of the Abstract of the first submitted manuscript as follows. We have underlined the parts we have modified in response to your comments.

Lines 24 to 37 of the revised manuscript:

Abstract: The Japanese government has implemented a new screening program to promote measures to avoid worsening lifestyle-related diseases and frailty among the older population. In this effort, the government formulated a new health assessment questionnaire for the screening program of old-old adults aged ≥75 years. The questionnaire comprises 15 items, of which 12 address frailty, two address general health status, and one addresses smoking habits. This study examined the construct validity of this questionnaire using the explanatory factor analysis (EFA) and confirmatory factor analysis (CFA). The data used in this study were drawn from a mail-in survey conducted in 2020 as part of the Septuagenarians, Octogenarians, Nonagenarians Investigation with Centenarians study. A total of 1,576 respondents (range, 78–99 years of age) were included in the study. Although the EFA did not show an interpretable factor structure of the questionnaire with 15 items, the CFA using only 12 frailty-related items showed the goodness of fit for a higher-order factor “frailty” and the five frailty-related sub-factors model was acceptable. These results suggest that the total score of the 12 frailty-related items in the questionnaire can be used as an indicator of the degree of “frailty.”

In addition, we have revised lines 270 to 276 of the Conclusions of the first submitted manuscript as follows. We have underlined the parts we have revised in response to your comments.

Lines 230 to 236 of the revised manuscript:

In contrast, the results from the CFA showed that there were five sub-factors: physical function, nutritional status, oral function, cognitive function, and social aspects, indicating the goodness of fit of the model with the higher-order factor of "frailty" on top of these five sub-factors. In addition, the goodness of fit of the model that assumes frailty as the secondary factor may indicate that the total score of the 12 frailty-related items in the health assessment questionnaire for old-old adults can be used as an indicator of the degree of "frailty".

Lines 295 to 296 of the revised manuscript:

We propose that the total score of the 12 frailty-related items in the health assessment questionnaire for old-old adults can be used as an indicator of the degree of "frailty."

Comments to the Authors #2

Discussion and comparison with other countries' questionnaires about similar topics are very short (223-224 lines), please prolong it.

Response

Thank you for your suggestions. In response to your comments, we have added a description of our discussion of frailty assessment tools in Japan and abroad.

Lines 241 to 256 of the revised manuscript:

In addition to this questionnaire, only some comprehensively assessed the physical, psychological, and social aspects of frailty, such as the Groningen Frailty Indicator [21] [22] [23] from Germany, the Edmonton Frail Scale [24] [25] from Canada, the Tilburg Frailty Indicator [26] [27] [28] from The Netherlands, and the Kihon Checklist [29] [30] and Kaigo Yobo Checklist [31] from Japan. These comprehensive frailty assessment instruments have been reported to examine convergent validity and/or concurrent validity and to compare their predictive validity: a comparison of predictive validity for disability between the Groningen Frailty Indicator and the Tilburg Frailty Indicator [32]; a comparison of predictive value for identifying frailty between the Groningen Frailty Indicator and the Tilburg Frailty Indicator [33]; an examination of the convergent validity between the electronic frailty index and the Edmonton Frail Scale [34]; a comparison of diagnostic test accuracy for identifying frailty among the Groningen Frailty Indicator, the Edmonton Frail Scale, and the Kihon Checklist [35]; a comparison of predictive value for frailty between the Kihon Checklist and the Kaigo Yobo Check-list [36]; an examination of convergent validity between the Kihon Checklist (Portuguese version) and the Edmonton Frail Scale [37].

Comments to the Authors #3

What are the strengths of the study?

Response

Thank you for your comment. We have added the following sentence to the Conclusions section to describe the strengths of our manuscript clearly.

Lines 296 to 300 of the revised manuscript:

We expect that health professionals working in municipalities will be able to use the total score of the 12 frailty-related items out of the 15 items of the questionnaire to identify those who are eligible for health guidance aimed at preventing frailty among community-dwelling older adults.

Comments to the Authors #4

The smaller part of the literature sources are from the last 5 years, and only 5 are from different countries than Japan, sometimes a year is not in bold, in the last source (349 line) year is missing...

Response

Thank you for your suggestion. We have added 15 references reported from countries other than Japan to our revised manuscript. Of the 15 references we added, nine have been reported during the past five years (2017-2022). We have bolded the publishing year for all references, rather than website.

The references added in the revised manuscript:

  1. Pett, M.A., Lackey, N.R., Sullivan, J.J. Making Sense of Factor Analysis. SAGE: Thousand Oaks, United States, 2003, ISBN 0-7619-1950-3.
  2. de Vries, N.M.; Staal, J.B.; van Ravensberg, C.D.; Hobbelen, J.S.; Olde Rikkert, M.G.; Nijhuis-van der Sanden, M.W. Outcome instruments to measure frailty: a systematic review. Ageing Res Rev. 2011, 10, 104-114.
  3. Sezgin, D.; O'Donovan, M.; Cornally, N.; Liew, A.; O'Caoimh, R. Defining frailty for healthcare practice and research: A qualitative systematic review with thematic analysis. Int J Nurs Stud. 2019, 92, 16-26.
  4. Gabrovec, B.; Veninšek, G.; Samaniego, L.L.; Carriazo, A.M.; Antoniadou, E.; Jelenc, M. The role of nutrition in ageing: A narrative review from the perspective of the European joint action on frailty - ADVANTAGE JA. Eur J Intern Med. 2018, 56, 26-32.
  5. Parisius, K.G.H.; Wartewig, E.; Schoonmade, L.J.; Aarab, G.; Gobbens, R.; Lobbezoo, F. Oral frailty dissected and conceptual-ized: A scoping review. Arch Gerontol Geriatr. 2022,100, 104653.

  1. Peters, L.L.; Boter, H.; Burgerhof, J.G.; Slaets, J.P.; Buskens, E. Construct validity of the Groningen Frailty Indicator established in a large sample of home-dwelling elderly persons: Evidence of stability across age and gender. Exp Gerontol. 2015; 69, 129-141.
  2. Rolfson, D.B.; Majumdar, S.R.; Tsuyuki, R.T.; Tahir, A.; Rockwood, K. Validity and reliability of the Edmonton Frail Scale. Age Ageing. 2006, 35, 526-529.
  3. Perna, S.; Francis, M.D.; Bologna, C.; Moncaglieri, F.; Riva, A.; Morazzoni, P.; Allegrini, P.; Isu, A.; Vigo, B.; Guerriero, F.; Rondanelli, M. Performance of Edmonton Frail Scale on frailty assessment: its association with multi-dimensional geriatric conditions assessed with specific screening tools. BMC Geriatr. 2017, 17, 2.

  1. Zhang, X.; Tan, S.S.; Bilajac, L.; Alhambra-Borrás, T.; Garcés-Ferrer, J.; Verma, A.; Koppelaar, E.; Markaki, A.; Mattace-Raso, F.; Franse, C.B.; Raat, H. Reliability and Validity of the Tilburg Frailty Indicator in 5 European Countries. JAMDA. 2020, 21, 772-779.
  2. Zamora-Sánchez, J.J.; Urpí-Fernández, A.M.; Sastre-Rus, M.; Lumillo-Gutiérrez, I.; Gea-Caballero, V.; Jodar-Fernández, L.; Julián-Rochina, I.; Zabaleta-Del-Olmo, E. The Tilburg Frailty Indicator: A psychometric systematic review. Ageing Res Rev. 2022, 76, 101588.

  1. Daniels, R.; van Rossum, E.; Beurskens, A.; van den Heuvel, W.; de Witte, L. The predictive validity of three self-report screening instruments for identifying frail older people in the community. BMC Public Health. 2012, 12, 69.
  2. Si, H.; Jin, Y.; Qiao, X.; Tian, X.; Liu, X.; Wang, C. Comparing diagnostic properties of the FRAIL-NH Scale and 4 frailty screening instruments among Chinese institutionalized older adults. J Nutr Health Aging. 2020, 24, 188-193.
  3. Brundle, C.; Heaven, A.; Brown, L.; Teale, E.; Young, J.; West, R.; Clegg, A. Convergent validity of the electronic frailty index. Age Ageing. 2019, 48, 152-156.
  4. Ambagtsheer, R.C.; Visvanathan, R.; Dent, E.; Yu, S.; Schultz, T.J.; Beilby, J. Commonly used screening instruments to identify frailty among community-dwelling older people in a general practice (primary care) setting: a study of diagnostic test accuracy. J Gerontol A Biol Sci Med Sci. 2020, 75, 1134-1142.

  1. Sewo Sampaio, P.Y.; Sampaio, R.A.; Yamada, M.; Ogita, M.; Arai, H. Validation and translation of the Kihon Checklist (frailty index) into Brazilian Portuguese. Geriatr Gerontol Int. 2014, 14, 561-569.

Reviewer 2 Report

-Increase the number of references, they are few.
-It is not understood why eigenvalues are used, what is their contribution to the research.
- The relevance of the chosen methods is not clear to me. It is recommended to better explain Figure 2.
- Why is it assured in the discussion that it is statistically independent?
- It would be interesting to add what are the impacts of the results, how it influences government programs and communities.

-It is advisable to add more research findings

Author Response

Responses to the Reviewer 2

 First of all, we appreciate your important and helpful comments on our manuscript. We have indicated in the red text where we have made revisions and underlined where we had made revisions in response to your comments/suggestions. Apart from the comments from the two reviewers, values of the estimated communality in Table 4 and the order of the subfactors in Figure 2 have been revised.

Comments to the Authors #1

-Increase the number of references, they are few.

Response

Thank you for your comment. As a result, the number of cited references in the revised manuscript is now 39.

Comments to the Authors #2

-It is not understood why eigenvalues are used, what is their contribution to the research.

Response

Thank you for your comment. There are at least two tests for determining the number of factors in exploratory factor analysis: the Kaiser-Guttman test, in which factors with eigenvalues greater than 1 are employed, and the Scree test, in which the magnitude of the eigenvalues is plotted and extracted until before the transition becomes gradual. We used these two tests. To help the reviewer and readers better understand eigenvalues in the explanatory factor analysis, we have added descriptions of the use of eigenvalues in the exploratory factor analysis to the Methods and the Results sections.

Lines 135 to 137 of the revised manuscript:

We used both the Kaiser-Guttman test and the scree test to determine the number of factors to be extracted in the EFA [14].

Lines 178 to 181 of the revised manuscript:

Based on the Kaiser-Guttman test, we extracted five factors with eigenvalues of one or more. Next, the eigenvalues of the identified factors were plotted on a scree plot (Figure 1). Because the shape of the scree plot showed a smooth decrease after the fifth factor, we determined that there were five factors in this EFA.

Comments to the Authors #3

- The relevance of the chosen methods is not clear to me. It is recommended to better explain Figure 2.

Response

Thank you for your recommendation. To help the reviewer and readers better understand the confirmatory factor analysis, we have added substantially to the description of the method of the confirmatory factor analysis and the interpretation of the results (Figure 2) in the revised manuscript. We drew Figure 2 so that the order of the subfactors in the comprehensive frailty assessment model (physical function, nutritional status, oral function, cognitive function, and social aspects) used in the confirmatory factor analysis is the same as the order of the five frailty-related domains shown in the figure.

Lines 138 to 150 of the revised manuscript:

Because the health assessment questionnaire for old-old adults includes 12 items to assess multidimensional aspects of frailty, such as physical, cognitive, and social aspects, we then used CFA to examine whether the 12 frailty-related items in this questionnaire fit the multidimensional model of frailty [15]. Recent review articles have described that an assessment tool for frailty should include physical, psychological, cognitive, and social dimensions [16] [17], and the physical domain includes three fac-tors: physical function, nutritional status [15], and oral health status [18]. We assume that, behind these five frailty-related domains, there is a concept of multidimensional frailty that affects each of the five domains. Therefore, we assumed that the model of frailty for this CFA had five sub-factors as the primary factors: physical function (Q7, Q8, Q9, and Q13), nutritional status (Q3 and Q6), oral function (Q4 and Q5), cognitive function (Q10 and Q11), and social aspects (Q14 and Q15), with the secondary factor of "a comprehensive concept of frailty" on top of these five sub-factors (Figure 2).

Lines 204 to 210 of the revised manuscript:

The next step was the CFA using only 12 frailty-related items. These items were developed as representatives of five frailty-related domains: physical function (Q7, Q8, Q9, and Q13), nutritional status (Q3 and Q6), oral function (Q4 and Q5), cognitive function (Q10 and Q11), and social aspects (Q14 and Q15). CFA assumed that the model had a higher-order factor (frailty) above these five frailty sub-factors. Figure 2 shows that all estimated factor loadings on the 12 items were 0.34 or more except for one item (Q 15), and all of them were statistically significant (P < 0.05).

Lines 230 to 236 of the revised manuscript:

In contrast, the results from the CFA showed that there were five sub-factors: physical function, nutritional status, oral function, cognitive function, and social aspects, indicating the goodness of fit of the model with the higher-order factor of "frailty" on top of these five sub-factors. In addition, the goodness of fit of the model that assumes frailty as the secondary factor may indicate that the total score of the 12 frailty-related items in the health assessment questionnaire for old-old adults can be used as an indicator of the degree of "frailty".

Comments to the Authors #4

- Why is it assured in the discussion that it is statistically independent?

Response

Thank you for your comment. To avoid misunderstanding by the reviewer and readers, we have deleted the "statistically independent" description in the revised manuscript and revised it as follows.

Lines 34 to 37 of the revised manuscript:

…, the CFA using only 12 frailty-related items showed the goodness of fit for a higher-order factor “frailty” and the five frailty-related sub-factors model was acceptable. These results suggest that the total score of the 12 frailty-related items in the questionnaire can be used as an indicator of the degree of “frailty.”

Lines 230 to 236 of the revised manuscript:

…, the results from the CFA showed that there were five sub-factors: physical function, nutritional status, oral function, cognitive function, and social aspects, indicating the goodness of fit of the model with the higher-order factor of "frailty" on top of these five sub-factors. In addition, the goodness of fit of the model that assumes frailty as the secondary factor may indicate that the total score of the 12 frailty-related items in the health assessment questionnaire for old-old adults can be used as an indicator of the degree of "frailty".

Comments to the Authors #5

- It would be interesting to add what are the impacts of the results, how it influences government programs and communities.

Response

Thank you for your suggestion. We have added the following sentence to the Conclusions section to describe the impacts of the results clearly.

Lines 296 to 300 of the revised manuscript:

We expect that health professionals working in municipalities will be able to use the total score of the 12 frailty-related items out of the 15 items of the questionnaire to identify those who are eligible for health guidance aimed at preventing frailty among community-dwelling older adults.

Comments to the Authors #6

-It is advisable to add more research findings. We have added the following sentences to the Results section to describe the results of the explanatory factor analysis and confirmatory factor analysis.

Response

Thank you for your suggestions. We have added the following sentences to the Results section.

Lines 184 to 199 of the revised manuscript:

The proportion of variance accounted by these five factors was 25.1%. The principal factor method was used to identify the factor loadings of each item by setting the number of factors to five. Promax rotation was performed to obtain the factor pattern matrix. Table 4 shows the factor loading of each item after factor rotation, and the estimated communality after factor identification. Factor loadings were interpretable with respect to the items of both Factor 1 (general health assessment: self-rated health status, life satisfaction) and Factor 4 (cognitive function: always asking the same thing, not remembering today’s date). However, the other factors were difficult to interpret because Factor 3 included various aspects, such as oral function, weight loss, and physical function, and Factor 5 included social relationships, nutritional status, and smoking habits. In addition, Factor 2 was difficult to interpret because there was only one item with a factor loading of ≥0.4. Although the estimated communality of items Q1 and Q2 were the highest, at 0.72 and 0.40, respectively, the communality of eight items was approximately 0.2. Specifically, the estimated communality of item Q12 "smoking habits " was the lowest (0.02). These results indicate that the questionnaire with 15 items was not explained by the five factors, and the EFA using 15 items did not extract interpretable factor structures.
